# Hierarchical Hyperbolic Embedding: Enhancing Long Context Comprehension of Large Language Models via Fusing Local and Global Features in Hyperbolic Space

## Abstract

Extending the context length of Large Language Models (LLMs) by improving Rotary Position Embedding (RoPE) has become a trend. While existing works mainly address RoPEs limitations within attention mechanism, this paper provides an analysis across nearly all parts of LLMs, uncovering their adverse effects on length generalization for RoPE-based attention. RoPE mapping knowledge to polar coordinates has the following drawbacks: 1) it cannot support exponential growth 2) it can not represent the complex hierarchical knowledge contained in the long context, 3)coordinates lead to distortions, in polar coordinates, points on different radii are ignored if they have the same Angle. Building on our observations, we propose HHPE(Hierarchical Hyperbolic Positional Embedding), which enhances long context comprehension of Large Language Models via fusing local and global features in hyperbolic space. HHPE can naturally represent hierarchical structure with equidistant embedding capability and exponential growth. Experiments across various model scales show that HHPE can maintain a more stable perplexity. Several analyses and ablations bring further support to our method and theoretical modeling.

## 1 Introduction

Generation based on the information from long contexts is crucial for Language Models (LLMs). However, LLMs are typically trained on a fixed context window Vaswani et al. (2017)Touvron et al. (2023)Groeneveld et al. (2024) and tends to overfit to the specific context length.

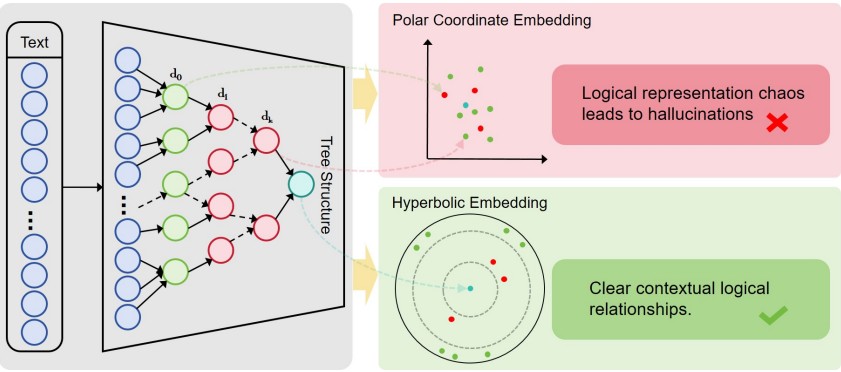

Figure 1: Long texts often contain tree-like logical structures, and embedding the knowledge of tree stumbles into polar coordinates will result in a failed embedding representation. From the figure, it can be seen intuitively that the tree-like logical structure represents chaos after embedding into polar coordinates, and embedding into hyperbolic space can clearly represent this relationship.

Many studies consider the absolute position embedding Vaswani et al. (2017)to be the source of over-fitting in long context generalization. As mitigation, several relative position embedding methods have been proposed Jin et al. (2024)Peng et al. (2024)Su et al. (2024)Press et al. (2022) to improve LLMs long-distance dependency. Among these, ALiBi Press et al. (2022) introduced a position-biased attention mask, which linearly declines the attention weights based on distance. ALiBi delivers stable perplexity in pre-training, but it loses the information from long-distance tokens, resulting in poor performance on long-context downstream tasks. Another method, RoPE Su et al. (2024) uses the phase of complex numbers to store the position information. Combined with continual pre-training and other interpolation-based methods. RoPE provides better access to long-distance information, making it one of the most widely used position embedding. However, RoPE-based LMs still struggle with long context generalization without supplementary methods.

In this paper, we take a closer look at the effect of ROPE in terms of hierarchical embedding representations. RoPE mapping knowledge to polar coordinates has the following drawbacks: 1) it cannot support exponential growth 2) it can not represent the complex hierarchical knowledge contained in the long context, 3)coordinates lead to distortions.

Building on our observations, we propose HHPE(Hierarchical Hyperbolic Positional Embedding), which enhancing long context comprehension of Large Language Models via fusing local and global features in hyperbolic space. Because (1)Negative curvature enables the hierarchy to scale exponentially. (2)Hyperbolic metrics can better reflect the relationship between hierarchies. (3) Suitable for semantic web, knowledge graph, social network and other tasks. HHPE can naturally represent hierarchical structure with equidistant embedding capability and exponential growth.

In summary, the contributions of this work are as follows:

- A novel method that enhancing long context comprehension of LLMS via fusing local and global features in hyperbolic space which solves the challenge of hierarchical knowledge representation.

- We begin by embedding the long text into Euclidean space, incorporating both token and position embeddings to construct a comprehensive embedding representation. Subsequently, this representation is mapped into hyperbolic space, where we extract long- and short-distance features based on the distance between the embedded representation and the origin. Finally, a gated mechanism is employed to fuse these features, yielding the final embedding representation.

- Comprehensive validation and evaluation of our method was presented. Detailed analysis of the results was offered and HHPE is compared with other approaches. Extensive experiments conducted on the Longbench V2 benchmark Bai et al. (2024b) demonstrate that our method achieves competitive performance in terms of quality and accuracy, notably achieving a best score.

In the second section, the study of hyperbolic space embedding and our highlights are analyzed. The third section analyzes the difference between polar coordinates and hyperbolic space for hierarchical knowledge embedding representation. The fourth section provides a detailed description of our proposed method. The fifth section presents experimental validation, demonstrating the effectiveness of our method and the impact of its individual components. Finally in the section conclusion, the paper concludes with a summary of our findings and contributions.

## 1.1 RELATED WORK

Hyperbolic embedding has gained significant attention in recent years due to its effectiveness in capturing hierarchical structures and complex relationships in data. A foundational study by Nickel & Kiela (2017) introduced Poincaré embeddings, demonstrating that hyperbolic space allows for efficient representations of hierarchical data while achieving superior performance over Euclidean embeddings in tasks like lexical entailment. Following this, Ganea et al. (2018) extended the approach by developing hyperbolic neural networks, which enable deep learning models to operate in hyperbolic space. Their work was later improved by Hyperbolic Graph Convolutional Networks (HGCN), proposed by Chami et al. (2019), which successfully generalized graph convolutional networks to hyperbolic space. Beyond graph structures, hyperbolic embeddings have been utilized in various domains. For instance, Tifrea et al. (2019) applied hyperbolic embeddings to word represen-

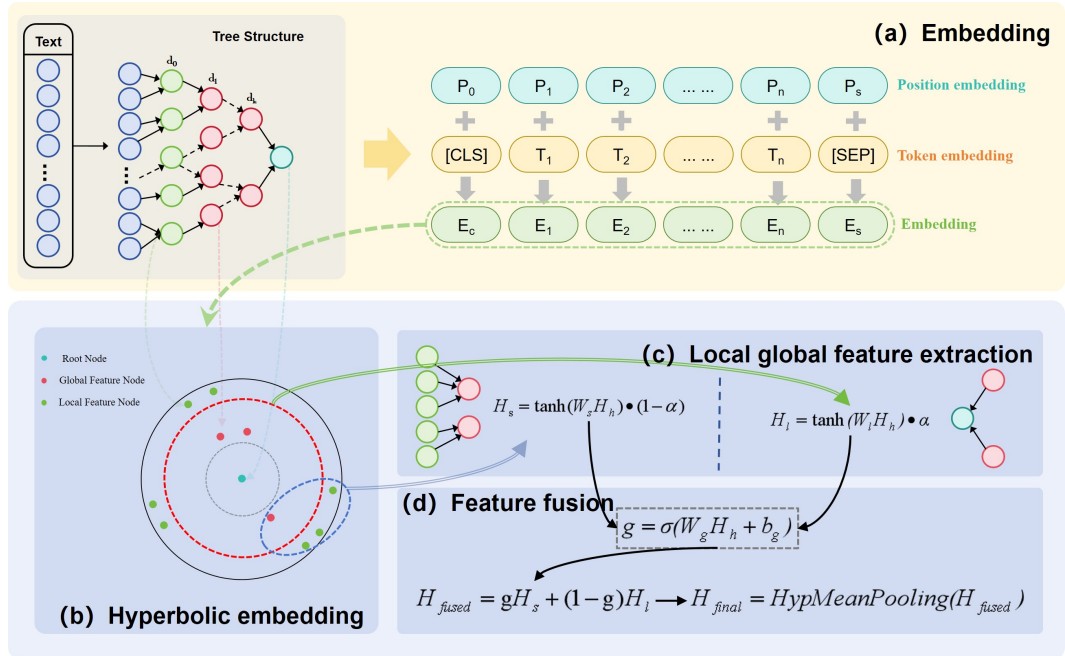

Figure 2: (a) We first embed the long text in Euclidean space, add the token and position embeddings to obtain a complete embedding representation, (b) and then embed the embedding representation into hyperbolic space, (c) extract long and short distance features according to the distance between the embedding result and the origin. (d) finally, we use a gated method to fuse long and short distance features to obtain the final embedding representation.

tations, showing improvements in hierarchical word similarity tasks. Additionally, Mathieu et al. (2019) proposed hyperbolic variational autoencoders (HVAE), leveraging hyperbolic geometry for generative modeling. Similarly, Yue et al. (2023) explored hyperbolic contrastive learning, enhancing the ability of models to learn discriminative representations in a self-supervised manner. Recent advancements have further refined hyperbolic embeddings for large-scale applications.Li et al. (2024) introduced hyperbolic attention networks, integrating hyperbolic geometry with attention mechanisms to improve sequence modeling. Overall, hyperbolic embedding techniques continue to evolve, with ongoing research focusing on scalability, optimization challenges, and better integration with deep learning architectures. To the best of our knowledge, HHPE is the first study to perform feature fusion in hyperbolic space embeddings

## 2 PRELIMINARIES

### 2.1 ANALYSIS OF THE LIMITATIONS OF POLAR COORDINATES

Polar Coordinates represent the location of a point using:

$$(x, y) = (r \cos \theta, r \sin \theta), \tag{1}$$

Where: $r$ is the distance from the origin (radial coordinates). $\theta$ is the Angle from the polar axis (angular coordinates). Hierarchical embeddings that cannot be represented well through Polar Coordinates embedding.

While polar coordinates provide an effective way to describe points in a two-dimensional space, they are inherently limited in capturing complex hierarchical structures. Specifically, hierarchical embeddings, which involve multi-level relationships and nested structures, cannot be effectively represented using a simple polar coordinate system. This limitation arises because polar coordinates primarily encode spatial relationships based on radial distance and angular displacement, without

an intrinsic mechanism to reflect hierarchical depth, parent-child dependencies, or multi-scale representations that are essential in hierarchical embeddings. Consequently, alternative geometric representations, such as hyperbolic embeddings, are often preferred for modeling hierarchical data due to their ability to naturally preserve tree-like structures and exponential growth properties.

### 2.1.1 EXPONENTIAL GROWTH

In hierarchical data such as semantic hierarchy, tree structure, the number of nodes in each level usually follows an exponential growth.

$$N_l = C^l \tag{2}$$

Where: $N_l$ denotes the number of nodes in the layer. $C$ is the number of children of each parent node (branching factor). $l$ indicates the depth of the hierarchy. In polar coordinates, the angular coordinate $\theta$ is limited to $[0, 2\pi]$:

$$\theta_i = \frac{2\pi i}{N^l} \left\{ \text{ i} = 1, 2, ..., N^l \right\} \tag{3}$$

The Angle has a limited range of values and cannot support an exponentially growing hierarchy. When the number of levels increases, the Angle separation between adjacent nodes tends to zero, which limits the representation ability.

### 2.1.2 HIERARCHICAL KNOWLEDGE REPRESENTATION

It can not represent the complex hierarchical knowledge contained in the long context. In a hierarchical embedding such as the WordNet semantic tree, we usually want to: 1. The high-level nodes are far from the origin and the low-level nodes are close to the origin. 2. The nodes in the same layer are equidistant from each other. However, polar coordinates fail to meet these requirements:

$$rl = a^l, \theta_i = \frac{2\pi i}{C^l} \tag{4}$$

Among them $a$ is the level growth factor, $c$ is the branching factor. When $l$ increases, the Angle interval will shrink rapidly:

$$\Delta\theta = \frac{2\pi}{C^l} \tag{5}$$

This causes neighboring nodes to almost overlap in Angle and cannot correctly represent the distinction between different branches.

### 2.1.3 DISTORTIONS

Coordinates lead to distortions, in polar coordinates, points on different radii are ignored if they have the same Angle . The Euclidean metric is used for polar coordinates:

$$d((r_1, \theta_1), (r_2, \theta_2)) = \sqrt{r_1^2 + r_2^2 - 2r_1 r_2 \cos(\theta_1 - \theta_2)} \tag{6}$$

When $r_1, r_2$ is vary large,The Angle difference $\cos(\theta_1 - \theta_2)$ may dominate the metric lead to

$$d \approx r_1 + r_2 \tag{7}$$

In this way, the distance between the levels increases linearly with $r$, while the true level relationship usually grows exponentially.

## 2.2 HYPERBOLIC SPACES

Hyperbolic Spaces (such as Poincare Balls) are more suitable for hierarchical embedding because:

### 2.2.1 SCALE EXPONENTIALLY

Negative curvature enables the hierarchy to scale exponentially.In hyperbolic Spaces (such as Poincare Balls), hierarchical relations follow the negative curvature measure:

$$\text{d}_B(x, y) = \log(1 + ||x + y||) \tag{8}$$

Regions far from the origin can accommodate exponentially growing nodes, avoiding limited representation power.

### 2.2.2 RELATIONSHIP BETWEEN HIERARCHIES

In hyperbolic space, due to the nature of negative curvature, the interval between each level grows exponentially, which can better separate the different levels, This allows for reasonable spacing between nodes even as the level increases.

### 2.2.3 APPLICABLE TO SEMANTIC WEB

In hyperbolic space, hyperbolic metrics ensure an exponential growth relationship between levels. In this way, the discriminative power of the embedding remains stable even when the level becomes deeper. Efficient Distance Encoding. It allows for exponentially increasing capacity with distance, effectively modeling relationships in large-scale graphs and networks. Low-Dimensional Expressiveness. Compared to Euclidean space, hyperbolic embeddings can preserve complex structures with fewer dimensions, making them computationally efficient. Scalability for Large Datasets. Due to its ability to compactly embed large graphs, hyperbolic space is beneficial for handling large-scale real-world networks and semantic structures.

## 3 HIERARCHICAL HYPERBOLIC POSITIONAL EMBEDDING(HHPE)

To mitigate the negative affect of polar position encoding in LLMs, we propose HHPE (Hierarchical Hyperbolic Positional Embedding) to modify hierarchical hyperbolic position encoding.Given a length of $l$, The input sequence of $X = [x_1, x_2, ..., x_n]$,We use hyperbolic embedding and multi-fusion network (MFN) for long and short term feature embedding and feature fusion.

### 3.1 HYPERBOLIC POSITIONAL EMBEDDING(HPE)

For the input word index $X$, we first get the word vectors:

$$W = Embedding(X) \in R^{l \times d} \tag{9}$$

Where, $d$ is the embedding dimension. Position encoding is performed on the sequence position index $P$ as follows.

$$P' = Embedding(P) \in R^{l \times d} \tag{10}$$

$$H = W + P' \tag{11}$$

Then map it to hyperbolic space:

$$H_h = \exp map_0(H) = \tanh(||H||)\frac{H}{||H||} \tag{12}$$

Among them, $\exp map_0(.)$ is the Poincare Ball exponential map.

### 3.2 HIERARCHICAL FEATURES EXTRACTION(HFE)

Calculate the hyperbolic distance from each point to the origin:

$$d_H(H_h0) = 2\tanh^{-1}(||H_h||) \tag{13}$$

Then the normalization operation is applied to it.

$$\alpha = \frac{d_H(H_h0) - d_{\min}}{d_{\max} - d_{\min}} \tag{14}$$

$\alpha$ ranges from $[0, 1]$ and represents a local (near 1) or global (near 0) weight.

$W_s$(Short-time projection matrix) focuses on local information, similar to low-order filtering (capturing local patterns at the n-gram level). $W_l$(long-term projection matrix) focuses on the global context, similar to higher-order transformations (capture long-distance relationships)

$$H_s = \tanh(W_s H_h) \bullet (1 - \alpha) \tag{15}$$

$$H_l = \tanh(W_l H_h) \bullet \alpha \tag{16}$$

Where $W_s$, $W_l$ is trainable parameter and $H_h$ is an embedding in hyperbolic space.

To reduce the computational cost, we can make both matrices share the same underlying transformation:

$$W_s = W + \Delta W_s \tag{17}$$

$$W_l = W + \Delta W_l \tag{18}$$

$W$ is the underlying projection matrix and , $\Delta W_s$,$\Delta W_l$ is the small adjustment term.In this way, $W_s$, $W_l$ shares most of the weights and improves the generalization ability and reduce the number of parameters to avoid overfitting.

## 3.3 FUSING LOCAL AND GLOBAL FEATURES (FLGF)

Traditional fusing local and global features:

$$H_{fused} = W_f [H_s, H_l] + b_f \tag{19}$$

Where $\Delta W_f$ is a trainable parameter of the fusion layer.

Here the Gating Mechanism is used, Calculate the gating coefficient, $g$ is used as the gating weight to determine the fusion ratio between $H_s$ and $H_l$,

$$g = \sigma(W_g H_h + b_g) \tag{20}$$

Where $W_g \in R^{d \times d}$ is the gating weight, $\sigma(.)$ is a Sigmoid function, $g$ guaranteed The value of g is between (0,1).

Make use of to smooth weighted short-term long-term features:

$$H_{fused} = \mathbf{g}H_s + (1 - \mathbf{g})H_l \tag{21}$$

When $g \approx 1$ short-time features contribute more, which is suitable for locally dependent tasks (e.g., n-gram). When $g \approx 0$ long-time features contribute more, which is suitable for remote dependent tasks (e.g., long text modeling)).

The fused $H_{fused}$ is still a variable-length sequence representation, but we need a fixed-size text embedding. For this reason, Hyperbolic Mean Pooling is used:

$$H_{final} = HypMeanPooling(H_{fused}) \tag{22}$$

$$HypMeanPooling(H) = \exp map_0^{-1}(\frac{1}{L} \sum_{i=1}^{L} \log map_0(H_i)) \tag{23}$$

Ensure that $H_{final}$ remains in the hyperbolic space.

## 4 EXPERIMENTS

In this section, To demonstrate the effectiveness of HHPE as both a position embedding and an extrapolation method, we conduct experiments during fine-tuning on C4 datasets Raffel et al. (2020), Longbench V2 benchmark is used to test the effectiveness of HHPE. Additionally, we perform ablation studies to analyze the impact of hyperparameters on HHPE.

### 4.1 FINE-TUNING IMPLEMENTATION DETAILS

Firstly we construct the required dataset for fine-tuning based on C4 Realnewslike data which contains 137,99,838 training and 13863 test sets. Specifically, the state-of-the-art Deepseek v3 is used to summarize each piece of data into Json files for llama-factory fine-tuning. The detailed parameters of the c4 dataset are given in table 3.

The parameter Settings for llama-factory fine-tuning are shown below in table 1. All models that require fine-tuning are trained on the same 4 RTX 3090 GPUs with 24 GB of GPU memory.

Table 1: Evaluation results (%) on LongBench v2.

| Model | Overall | Difficult | | Length | | |
| --- | --- | --- | --- | --- | --- | --- |
| | | Easy | Hard | Short | Medium | Long |
| Glm-4-9b-chat | 30.2 | 30.7 | 29.9 | 33.9 | 29.8 | 25.0 |
| Llama-3.1-8B | 30.0 | 30.7 | 29.6 | 35.0 | 27.9 | 21.3 |
| Qwen-2.5-7B | 27.0 | 29.2 | 25.7 | 36.1 | 23.7 | 18.5 |
| Qwen-2.5-72B | 39.4 | 43.8 | 36.7 | 44.0 | 34.0 | 41.7 |
| Glm-4-9b-chat-hhpe | 33.2 | 33.7 | 32.9 | 35.9 | 31.8 | 27.8 |
| Llama-3.1-8B-hhpe | 34.0 | 34.7 | 33.6 | 36.0 | 29.9 | 26.3 |
| Qwen-2.5-7B-hhpe | 32.0 | 33.2 | 29.7 | 38.1 | 26.7 | 23.5 |
| Qwen-2.5-72B-hhpe | 43.5 | 46.7 | 38.9 | 45.0 | 38.2 | 43.4 |

Table 2: Ablation Study results (%) on LongBench v2.

| Model | Overall | Difficult | | Length | | |
| --- | --- | --- | --- | --- | --- | --- |
| | | Easy | Hard | Short | Medium | Long |
| Glm-4-9b-chat | 30.2 | 30.7 | 29.9 | 33.9 | 29.8 | 25.0 |
| Glm-4-9b-chat-{w.}HPE | 31.1 | 31.6 | 30.8 | 34.8 | 31.6 | 26.1 |
| Glm-4-9b-chat-{w.}HHPE | 33.3 | 33.8 | 32.7 | 36.0 | 31.9 | 27.8 |

## 4.2 BASELINES

When evaluating the performance of HHPE on varing LLMs, selecting strong and diverse baselines is essential to ensure a fair and meaningful comparison. Below is a structured introduction to the four models as baselines for natural language processing (NLP) tasks.

- GLM-4-9B-Chat is an open-source variant of the latest generation in the GLM-4 series of pre-trained models, developed and released by Zhipu AI. This model is designed to support a wide range of natural language understanding and generation tasks, offering strong capabilities in reasoning, dialogue comprehension, and contextual adaptability.

- Llama-3.1-8B-Instruct is part of a collection of multilingual large language models (LLMs) that have been pretrained and instruction-tuned to enhance their generative performance. This model is specifically optimized for following instructions across different languages, making it suitable for various interactive AI applications, including chat-based systems, content generation, and automated assistance.

- Qwen 2.5-7B-Instruct is one of the latest models in the Qwen series of large language models, designed for advanced natural language processing tasks. With improvements in efficiency and contextual awareness, this model provides strong performance in instruction-following and generative text applications.

- Qwen 2.5-72B-Instruct represents a significant advancement in the Qwen model series, featuring substantial improvements in model scale, multilingual support, and context-processing capabilities. With its increased parameter size and enhanced training methodology, this model excels in complex NLP tasks, including long-form text generation, multilingual dialogue systems, and domain-specific applications requiring deep contextual understanding.

## 4.3 EXPERIMENTAL RESULTS

The Longbench V2 benchmark is utilized to evaluate the efficacy of HHPE. They apply middle truncation as described in Bai et al. (2024a) for sequences re-election the model's context window length. Different from them, we adopt the whole truncation method to deal with this problem.

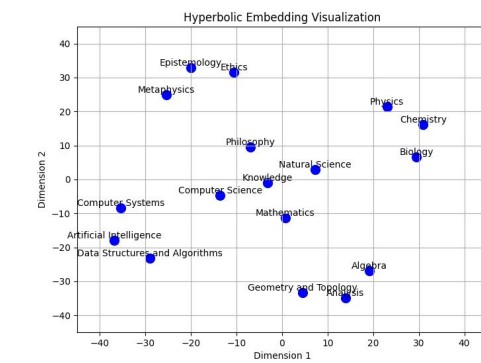

Figure 3: The representation after embedding in hyperbolic space is the result of a visualization whose dimensions are reduced to two dimensions using T-SNE.

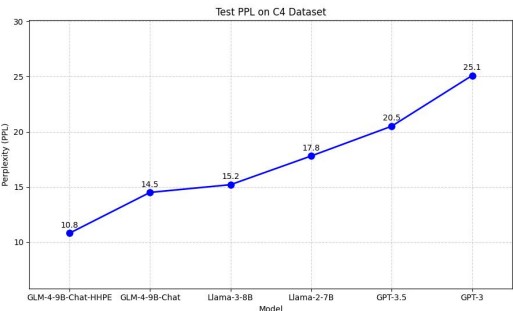

Figure 4: the GLM-4-9B-Chat-HHPE model demonstrates the best performance in this evaluation, achieving the lowest perplexity on the C4 dataset. The results highlight the effectiveness of hierarchical position encoding (HHPE) in further enhancing language model efficiency.

The following observations from table 1 can be summarized: In terms of effectiveness, our proposed model achieves promising results on all of the metrics compared with the baselines, which confirms the effectiveness of HHPE. These findings underscore the effectiveness and practical utility of HHPE, which holds the potential to enhance all RoPE-based open-source models.

## 4.4 ABLATION STUDY

Ablation experiments are conducted to showcase the validity of the components within the proposed method. We conduct ablation studies on various hyperparameters to observe their effects on our algorithm. Considering the consistent performance of HHPE across different parameter scales, we only evaluate the Glm-4-9b-chat models in ablation studies. The assessment outcomes are detailed in table 2 where w. means with. The results from table 2 clearly indicate that the overall model outperforms all other models in terms of performance. We have the following observation: (1) without the HFE and FLGF, the performance has decreased, which indicates the benefit of this module. (2) Notably, HPE module is shown to enhance performance by effectively represent the embedding of knowledge hierarchically.

## 4.5 CASE STUDY

In this case, we illustrate the process of extracting a hierarchical knowledge structure from unstructured text. The extracted knowledge representation is visualized as a tree-structured knowledge graph shown in Fig 5, where different domains of knowledge are organized in a logical hierarchy.

The root node represents the broadest concept, "Knowledge," which branches into major disciplines such as Natural Science, Mathematics, Computer Science, and Philosophy. Each of these domains is further subdivided into specialized subfields.

The figure 6 presents a hierarchical knowledge representation embedded within a Poincaré disk model, leveraging hyperbolic embeddings to effectively capture and preserve the hierarchical structure of concepts. In this visualization, the root node ("Knowledge") is positioned near the center of the Poincaré disk, with hierarchical subdomains branching outward. Each subsequent level in the hierarchy expands radially, with nodes placed closer to the boundary representing more specialized concepts. The hierarchical nature of the data is naturally encoded within the hyperbolic space, where distances between nodes reflect the underlying relationships between parent and child concepts.

The figure 3 presents the two-dimensional visualization of a hierarchical knowledge structure after being embedded in hyperbolic space and subsequently reduced using t-distributed Stochastic Neighbor Embedding (t-SNE). The purpose of this visualization is to preserve the relative distances and structural hierarchy inherent in the original high-dimensional hyperbolic embedding. The central nodes ("Knowledge," "Natural Science," "Mathematics," and "Computer Science") are positioned around the origin, indicating their higher-level status within the hierarchy. More specialized subfields (e.g., "Physics," "Algebra," "Artificial Intelligence") are distributed further away, reflecting their deeper placement in the original tree structure. Nodes with similar semantic or structural relationships (e.g., "Epistemology" and "Ethics" under "Philosophy") appear closer to each other, suggesting that t-SNE successfully preserved local similarities from the hyperbolic space.

### 4.6 PERPLEXITY COMPUTATION

Perplexity (PPL) is a standard metric for evaluating the performance of language models, where lower values indicate better predictive capabilities. This experiment assesses the PPL of various models on the C4 dataset, comparing their ability to generate coherent and fluent text.

The results, as shown in the figure, reveal distinct performance differences among the evaluated models: (1)GPT-3 and GPT-3.5 demonstrate the highest perplexity values, with 25.1 and 20.5, respectively. This indicates that these models exhibit relatively lower efficiency in predicting next-word probabilities on the C4 dataset. (2)Llama-2-7B and Llama-3-8B achieve significantly improved perplexity scores of 17.8 and 15.2, respectively, reflecting their enhanced optimization and architectural improvements. (3)GLM-4-9B-Chat outperforms the aforementioned models with a perplexity of 14.5, showcasing its efficiency in handling text generation tasks. (4)The GLM-4-9B-Chat-HHPE model achieves the lowest perplexity of 10.8, indicating superior performance in modeling language distributions and generating more predictable sequences. Overall, the GLM-4-9B-Chat-HHPE model demonstrates the best performance in this evaluation, achieving the lowest perplexity on the C4 dataset. The results highlight the effectiveness of hierarchical position encoding (HHPE) in further enhancing language model efficiency.

## 5 CONCLUSION

In this paper, we analyze RoPE-based positional embedding by mapping knowledge to polar coordinates has the drawbacks using the theory of manifolds. Our analysis reveals that HHPE achieves represent the embedding of knowledge hierarchically. We propose Hierarchical Hyperbolic Positional Embedding (HHPE) to enhances Experiments demonstrate that HHPE significantly improves long context comprehension of Large Language Models compared to baselines across diverse tasks on Longbench V2 benchmark. Our ablation studies and visualizations provide further support for our method and theoretical modeling.

## 6 LIMITATIONS

HHPE is effective for hierarchical knowledge modeling of long texts, but whether these theoretical assumptions are applicable to all types of tasks and data distribution, for example, short texts will increase computational overhead, the effect is not necessarily much better. How stable HHPE is in practical applications, such as legal, medical, code generation and other long text scenarios, remains to be investigated.

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

## A  APPENDIX

Table 3: Sizes for the C4 variants in english

| Name | Train | Validation |
|---|---|---|
| en | 364868892 | 364608 |
| en.noblocklist | 393391519 | 393226 |
| realnewslike | 13799838 | 13863 |

Table 4: Parameter Settings for llama-factory

| Parameter | Value |
|---|---|
| finetuning_type | lora |
| lora_rank | 8 |
| cutoff_len | 2048 |
| learning_rate | 1.0e-4 |
| lr_scheduler_type | cosine |
| warmup_ratio | 0.1 |
| bf16 | true |

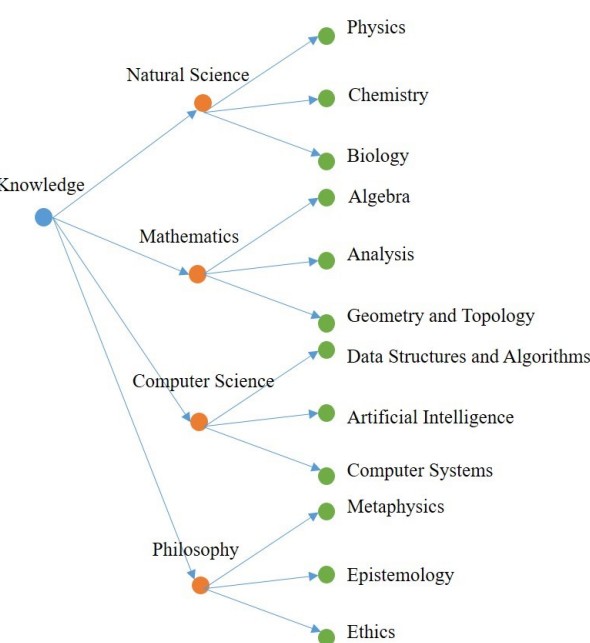

Figure 5: The tree-like knowledge extracted from long texts can be represented as a tree-like steel structure in a graph.

Figure 6: Results of visualization after embedding tree-like knowledge extracted from long texts into hyperbolic space.

