# OpenReview forum: "Hierarchical Hyperbolic Embedding: Enhancing Long Context Comprehension of Large Language Models via Fusing Local and Global Features in Hyperbolic Space"
_ICLR.cc/2026/Conference — ICLR 2026 Conference Withdrawn Submission_

### Official Review · Reviewer_YXLm · 2025-10-18

**Soundness:** 2
**Presentation:** 2
**Contribution:** 2
**Rating:** 2
**Confidence:** 4

**Summary:**

This paper proposes Hyperbolic Hierarchical Position Encoding (HHPE). The core idea is to use hyperbolic space, rather than the traditional polar coordinate system, for the LLM's positional encoding. This is intended to better represent hierarchical knowledge, support exponential growth, and prevent the distortion associated with RoPE. The results on LongBench V2 indicate that models equipped with HHPE generally outperform traditional models.

**Strengths:**

1.  The paper presents a novel perspective by innovatively using hyperbolic space to improve positional encoding.
2.  Experiments were conducted on a variety of different models.

**Weaknesses:**

1. The method's purpose is weird. In my view, the function of positional encoding is to represent positional information, whereas the semantic information of the token itself is not something positional encoding should be responsible for. Therefore, why you try to use a positional encoding to represent hierarchical knowledge?
2. HHPE consists of three distinct operations, but the paper does not clearly articulate the motivation behind their design. The rationale for choosing these specific operations is not well-explained.
3. The evaluation is limited to LongBench V2, lacking other general-purpose datasets. Given that HHPE introduces significant modifications to the model, its general capabilities should be more broadly assessed.
4. The use of a line chart in Figure 4 is inappropriate, as the data points do not have a sequential relationship. Furthermore, comparing Perplexity (PPL) across models of different sizes is not meaningful. The comparison should instead focus on the change in perplexity for the *same model* before and after applying HHPE.
5.  The paper fails to specify the engineering implementation details of how HHPE is applied to an LLM. For instance:
    * Is the model's original RoPE (Rotary Position Embedding) removed?
    * In HHPE, where does the initial position encoding come from?
    * Does the calculated final hidden state, $H_{final}$, replace the token's original hidden state?
6.  The paper does not clearly describe the specific fine-tuning data format. The C4 dataset is a pre-training corpus, so its use for fine-tuning is questionable. The paper should also explain why it was necessary to use DeepSeek to summarize each data sample into a JSON format.
7.  There are formatting errors, such as an incorrect table reference in Section 4.1.

**Questions:**

see weakness 1,2,5,6

---

### Official Review · Reviewer_Jxi7 · 2025-10-28

**Soundness:** 1
**Presentation:** 1
**Contribution:** 2
**Rating:** 0
**Confidence:** 4

**Summary:**

This paper proposes Hierarchical Hyperbolic Positional Embedding (HHPE), a new positional encoding method for LLMs.  the authors identify that RoPE’s mapping into polar coordinates leads to distortions, poor hierarchical representation, and limited scalability for long text contexts. To address this, HHPE utilyzes hyperbolic embeddings and introduces HPE and FLGF techniques. Experiments show consistent improvements with lower perplexity and better long-context performance.

**Strengths:**

The introduction of hierarchical structure information in positional encoding is intuitive and well-motivated. and the experimental results demonstrated by the authors show consistent performance improvements.

**Weaknesses:**

The manuscript is very poorly written, with many grammatical errors, incorrect punctuation, and mixed use of upper and lower case letters. All of which do not suit the standard for academic submissions.

The figures referenced in Section 4 are not presented in the main draft of the paper; instead, they are provided in the appendix.

A discussion should be provided on other recent studies, including [1].
[1] Dai, Chang, et al. "HoPE: Hyperbolic Rotary Positional Encoding for Stable Long-Range Dependency Modeling in Large Language Models." arXiv preprint arXiv:2509.05218 (2025).

**Questions:**

N/A

---

### Official Review · Reviewer_nntG · 2025-10-30

**Soundness:** 1
**Presentation:** 1
**Contribution:** 2
**Rating:** 2
**Confidence:** 4

**Summary:**

This paper proposes a novel positional embedding method, Hierarchical Hyperbolic Positional Embedding (HHPE), which maps embeddings into a hyperbolic space (Poincaré ball) to better model long-range and hierarchical dependencies in text. The method involves extracting local and global features based on hyperbolic distance and fusing them with a gated mechanism. The authors evaluate HHPE by fine-tuning several base LLMs (GLM, LLaMA, Qwen) and testing on the LongBench V2 benchmark and C4 perplexity, reporting improvements over the base models.

While the core idea of using hyperbolic geometry for positional encoding is novel and potentially significant, the paper suffers from critical flaws in its technical execution, experimental design, and presentation. These issues are severe enough to undermine the paper's claims and warrant rejection.

The idea of using hyperbolic space for positional encoding is promising. However, this paper in its current form does not provide a convincing case for HHPE. The experimental design probably has a fundamental confound that likely explains the reported results, the implementation details remmains vague, and the writing is not up to the standard required for a scientific publication. For these reasons, I recommend rejection.

Should the authors address these points in a rebuttal—particularly by justifying the experimental design and demonstrate the complete implementation details—the paper could be re-evaluated. However, the required revisions are substantial.

**Strengths:**

The core idea of leveraging hyperbolic geometry for positional encoding is innovative and theoretically well-founded. The argument that hyperbolic space is more naturally suited for representing hierarchical, tree-like structures (common in long-form text) than the polar coordinates underlying RoPE is compelling and represents a meaningful conceptual advance.

The inclusion of ablation studies (Table 2) helps to validate the contribution of individual components of the proposed HHPE architecture (HPE, HFE, FLGF), strengthening the empirical claims.

The use of t-SNE and Poincaré disk visualizations to illustrate the hierarchical structure of the learned embeddings is a strong point, providing intuitive, qualitative support for the theoretical claims.

**Weaknesses:**

The experimental design is flawed and probably invalidates the core claims. According to line 323, the paper very likely compares HHPE-augmented models *after fine-tuning* against baseline models *in their original, pre-trained state*. This introduces a massive confounding variable: the observed performance gains on LongBench V2 are likely attributable to the domain adaptation from fine-tuning on C4, not the HHPE method itself. Without comparing against baselines that have undergone the *identical fine-tuning procedure*, the central claim that HHPE is responsible for the improvement is unsupported. This is a fatal flaw in the empirical validation.

The design choices are lack of explicit motivation and are poorly justified, leaving the rationale behind them unclear. The proposed two-level feature extraction (local & global) can be an oversimplification that is not adequately justified, raising doubts about its capability to model the "exponentially growing" hierarchical knowledge structures that are the paper's central motivation. The connection between the single scalar gating mechanism and the representation of exponential, tree-like hierarchies remains underexplained, creating a gap between the method's implementation and its intended theoretical benefits.

The paper fails to demonstrate its key motivation: improved hierarchical modeling. The introduction and motivation heavily emphasize that HHPE is designed to capture hierarchical knowledge structures. However, the experiments only evaluate long-context comprehension and general perplexity. There is no specific evaluation or analysis that demonstrates HHPE's superiority in modeling hierarchical relationships (e.g., on tasks like parsing, logical reasoning over hierarchies, or QA on hierarchically structured documents). The paper does not provide evidence for the problem it claims to solve.

The methodology is unconvincing and the implementation details are vague. The technical description in Section 3 lacks rigor. The operations for projecting into and operating within hyperbolic space are poorly explained and not well-justified. The description of the "multi-fusion network" is vague. The integration of HHPE into the transformer architecture is ambiguous—it is unclear how HHPE, which seems to require separate word and positional embeddings, is integrated into models like LLaMA that use RoPE, which is fused with the query/key vectors. Was RoPE removed? If so, how was the model's existing capacity preserved? If not, was HHPE applied only after the embedding layer or applied each layer after applying RoPE? These omissions make the method irreproducible and its true architectural impact unclear.

The paper did insufficient comparison with other Positional Embeddings. The paper only compares against base models that apply RoPE as positional embedding. Despite of the fact that the authors only aimed at enhancing RoPE based models' ability, comparison with other positional embedding methods e.g. CoPE is essential to demonstrate that HHPE offers a distinct advantage.

The paper suffers from significant presentation and clarity issues. The paper is difficult to read due to numerous grammatical errors, typos, and awkward phrasing. Key sentences are confusing or ungrammatical. For instance, the abstract mentions an "analysis across nearly all parts of LLMs" that is not found in the paper; line 323 claims the finetuning parameter settings are shown in table 1 but in fact they are in table 4 in the appendix; and a mention of "sequence re-election" (line 376) is unexplained. In line 065 the paper uses "ROPE" to represent RoPE and in table 1 it uses "Glm" for GLM. This lack of polish severely hinders the understanding and assessment of the work.

**Questions:**

(1) Why you choose the proposed 2-level (local & global) architecture? A scalar gating method plus only 2-level feature extraction can be insufficient to model the hierarchical/tree structured context which is your main concern. Can you provide more rationales to justify its theoretial appropriateness? Recently, Chang et al. show that simply replacing RoPE's rotation with Lorenz rotation will effectively introduce hyperbolic features in the paper "HoPE: Hyperbolic Rotary Positional Encoding for Stable Long-Range Dependency Modeling in Large Language Models". I'm not intending to do comparasion, but can you give more clue to show the specific advantges of your design choices?

(2) Precisely how did you integrate HHPE into the transformer architecture of the baseline models? Did you remove their native RoPE? If so, how did you compensate for the change in model structure and initialization? If not, was HHPE applied only after the embedding layer or applied each layer after applying RoPE? A clear, detailed description is needed.

(3) The paper motivates HHPE based on hierarchical knowledge, but no such evaluation is presented. Can you provide experiments that directly test the model's ability to understand and reason over hierarchical structures?

(4) Can you provide results from the critical controlled experiment: comparing your HHPE models against the baseline models (GLM, LLaMA, Qwen) after they have been fine-tuned on the *exact same* C4 dataset with the *exact same* hyperparameters and setup?

(5) Equation 8 for the Poincaré distance is incorrect. What was the exact formula used in your implementation? Please also clarify how the parameters $d_{min}$ and $d_{max}$ in Eq. 14 are computed during training and inference.

---

### Official Review · Reviewer_xk93 · 2025-10-31

**Soundness:** 2
**Presentation:** 2
**Contribution:** 2
**Rating:** 2
**Confidence:** 4

**Summary:**

This paper proposes a hyperbolic embedding that attempts to embed a hierarchical "tree-like" structure of different tokens into a hyperbolic space. The paper claims that the polar coordinates are unable to embed hierarchical structures and proposes to assign two different weight matrices for "local patterns" and "long-distance relationships". To verify the effectiveness of the proposed method, experimental results are presented about various models fine-tuned on the C4 dataset and evaluated on Longbench.

**Strengths:**

* The proposed method is interesting, presenting a new design of embedding that may capture distinct "local" or "global" features of each token.

* The figures in the paper are clear, demonstrating the idea of the proposed method.

**Weaknesses:**

* The analyses provided in the paper about why hyperbolic space is better than polar coordinates are not convincing enough. The analyses are based on the distance between queries (or keys) of different tokens. However, the attention score is determined by the inner product of query and key. From this perspective, the superiority of the hyperbolic space is in question.

* While the paper presents some tree-like structures of semantic symbols in the appendix, there is a lack of analysis on how these semantic hierarchical structures are reflected in a token-level embedding. It is hard to imagine that some tokens are "roots" while others are "leaves".

* The details of the proposed method and experiments are missing.
   1. Eq. 10 mentions position encoding. Does that mean the proposed method uses absolute positional encoding instead of rotary position embedding? How does the proposed method apply to those models with RoPE? (For example, the paper presents results from finetuning the Llama and Qwen models.)

   2. Since it is an embedding, does each token have its corresponding $W_s$, $W_l$ and $W_g$ as mentioned in Section 3? Will it introduce too many additional parameters?

   3. The paper does not provide any details about how the experiments are actually conducted (like hyper-parameter setting, how the models are adapted to the proposed embedding).

   4. There is no time complexity analysis or details about the number of parameters. Since the proposed method may introduce additional parameters in the embedding, we can not determine whether the performance gain is due to the effectiveness of the proposed embedding or the increased number of additional parameters.

**Questions:**

My primary concern is with the unclear description of the proposed method and experiments. It would be appreciated if the authors could clarify how the proposed embedding is applied to pretrained models and how the experiments are conducted.

* How many additional parameters are introduced with the proposed method?

* Whether the proposed method uses absolute position encoding or RoPE?

* What are the hyperparameters when the experiments are conducted?

---

### Public Comment · ~Ermo_Hua1 · 2025-11-12
**Concerns regarding the potential plagiarism in submission 18739**

Dear Committee:

We are writing to express our concern regarding potential plagiarism in submission 18739 to ICLR 2026, titled "Hierarchical Hyperbolic Embedding: Enhancing Long Context Comprehension of Large Language Models via Fusing Local and Global Features in Hyperbolic Space." This submission appears to bear significant similarities to our ICML 2025 paper "Fourier Position Embedding: Enhancing Attention’s Periodic Extension for Length Generalization." However, there is not any citation of FoPE in the submission 18739, which may violate the academic norms.

The exactly matched parts includes:
- Abstract (line 14-29)
- Introduction (line 33-70)

The quite similar parts includes:
- Conclusion (line 472-478)

All of the listed lines of submission 18739 are either identical or strikingly similar to the text of the ICML 2025 paper. We trust that the committee will give this matter its due attention and would welcome your prompt response.

The Authors of FoPE,
Tsinghua University

---

### Note · Authors · 2025-11-24

I have read and agree with the venue's withdrawal policy on behalf of myself and my co-authors.